# Tissue-Resident Memory T Cells in Rheumatoid Immune Diseases: Pathogenic Mechanisms and Therapeutic Strategies

**DOI:** 10.3390/biomedicines13122945

**Published:** 2025-11-30

**Authors:** Yu Tian, Jie Zhang, Lianying Wu, Chi Zhang, Fan Zheng, Yang Yang, Guanting Lu, Daoyuan Xie

**Affiliations:** 1Laboratory of Translational Medicine Research, Deyang People’s Hospital of Chengdu University of Traditional Chinese Medicine, Deyang 611137, China; tianyu950715@gmail.com (Y.T.); 18380161308@163.com (L.W.); 2Department of Rheumatology and Immunology, Deyang People’s Hospital, Deyang 611137, China; jiezhangdoc@sina.com; 3Department of Genetics, MD Anderson Cancer Center, The University of Texas, Houston, TX 77030, USA; czhang11@mdanderson.org; 4School of Medical Sciences, University Sains Malaysia, Kota Bharu 16150, Malaysia; fanzheng123@student.usm.my; 5Department of Pathology, Deyang People’s Hospital, Deyang 611137, China; 15883840906@163.com; 6Department of Oncology, Deyang People’s Hospital of Chengdu University of Traditional Chinese Medicine, Deyang 611137, China

**Keywords:** tissue-resident memory T cells, rheumatoid immune diseases, pathogenic mechanisms, therapeutic strategies

## Abstract

Tissue-resident memory T (T_RM_) cells persist long-term in non-lymphoid tissues and provide rapid local immune protection, yet emerging evidence shows they also act as key drivers of chronic inflammation and relapse in rheumatoid immune diseases such as rheumatoid arthritis (RA), systemic lupus erythematosus (SLE), systemic sclerosis (SSc), and primary Sjögren’s syndrome (pSS). A systematic search of PubMed, Web of Science, and Google Scholar (through October 2025) identified studies on T_RM_ cell biology, pathogenic roles, and therapeutic modulation in autoimmune diseases. This review summarizes the fundamental features of T_RM_ cells, including their TGF-β and IL-15 dependent development, tissue-specific heterogeneity, and unique metabolic programs. It highlights disease-specific pathogenic mechanisms: promotion of osteoclastogenesis and chronic synovial inflammation via Granulocyte-macrophage colony stimulating factor (GM-CSF) and the IL-23/IL-17 axis in RA; amplification of type I interferon responses and autoantibody production in SLE; potential contribution to fibrosis through TGF-β secretion in SSc; and mediation of glandular injury through cytotoxicity in pSS. Therapeutic strategies targeting T_RM_ cells—such as JAK inhibitors, IL-17/IL-23 blockade, disruption of residency signals, metabolic interventions, and microenvironmental remodeling via nanotechnology—are critically evaluated. Challenges remain in achieving tissue-specific targeting without compromising systemic immune memory. Future directions include spatial transcriptomics, organoid models, and artificial intelligence to support precision medicine. Targeting T_RM_ cells presents a promising novel avenue for achieving long-term remission and potentially even a cure for rheumatoid immune diseases.

## 1. Introduction

Rheumatic and autoimmune diseases, such as rheumatoid arthritis (RA), psoriatic arthritis, and systemic lupus erythematosus (SLE), are characterized by immune-mediated dysregulation of stromal tissues, chronic inflammation, joint destruction, and aberrant immune mechanisms [1]. Therapeutic approaches have traditionally relied on non-steroidal anti-inflammatory drugs, glucocorticoids, and conventional disease-modifying antirheumatic drugs (DMARDs). In recent years, several innovative therapies have emerged, including biologics such as IL-6 inhibitors, small-molecule targeted agents like JAK inhibitors, cellular therapies including mesenchymal stem cells and CAR-T cells, as well as genome-editing technologies such as CRISPR-Cas9 [2,3,4,5,6]. Although these therapies provide meaningful clinical benefits by alleviating symptoms and modulating inflammatory pathways, frequent disease relapse and pathological rebound suggest that conventional approaches do not adequately control the underlying immune memory mechanisms [7,8]. Hence, overcoming this therapeutic limitation requires novel interventions targeting the fundamental logic of immune memory.

Tissue-resident memory T (T_RM_) cells are a subset of long-lived memory T cells that reside in non-lymphoid barrier tissues, exhibiting unique localization, functional, and metabolic adaptations [9,10]. Recent studies have shed new light on the mechanisms driving disease relapse in rheumatic autoimmune disorders. T_RM_ cells, which reside predominantly in barrier tissues such as the skin, lungs, gut, liver, and brain, persist long-term and are poised for rapid response to local stimuli. They secrete potent inflammatory mediators and play critical roles in both anti-tumor immunity and autoreactive immune responses [11,12,13,14]. Typical surface markers of T_RM_ cells include CD69, CD103, and CD49a, which collectively mediate tissue retention and functional activity [10,13,15]. CD69 suppresses S1P1-dependent tissue egress, CD103 binds epithelial E-cadherin to promote adhesion, and CD49a interacts with collagen to support basement-membrane localization and cytotoxic function [16,17].

The pathogenic role of T_RM_ cells in rheumatic autoimmune diseases is increasingly recognized. In psoriatic arthritis, CD69^+^CD103^+^CD8^+^ T_RM_ cells are enriched in synovial fluid and secrete high levels of IL-17A, indicating their involvement in local inflammation and disease relapse [18]; Similarly, in autoimmune hepatitis, hepatic CD8^+^ T_RM_ cell abundance correlates positively with disease severity [19]. In murine arthritis models and human RA synovium, CD8^+^ T_RM_ cells persist in previously inflamed sites during remission and recruit circulating effector cells via the chemokine (C-C motif) ligand 5 (CCL5) axis, triggering arthritis flare-ups—a phenomenon termed “lesion memory” [20]. Spatial transcriptomics and multiplex immunofluorescence analyses of RA synovial tissue have revealed dense clusters of CD8^+^CD69^+^CD103^+^ T_RM_ cells co-localized with high levels of IL-6 and TNF-α. These findings suggest that T_RM_ cells sustain persistent local inflammation through a “lesion memory”–driven mechanism [21]. Collectively, these findings highlight T_RM_ cells as central mediators of “lesion memory,” sustaining disease activity and relapse in rheumatic autoimmune disorders.

Current therapies predominantly target circulating immune cells and exert limited effects on tissue-resident T_RM_ cells. Consequently, developing strategies that specifically modulate T_RM_ cells—such as disrupting key survival pathways like IL-15 or TGF-β signaling, or blocking T_RM_-associated surface molecules—is essential for achieving durable remission and potentially curing rheumatic autoimmune diseases [22,23].

Based on this background, the present study aims to elucidate the pathogenic mechanisms of T_RM_ cells in the relapse of rheumatic autoimmune diseases and to explore their potential as therapeutic targets. By systematically analyzing T_RM_ cell phenotypes, distribution, and functions, this work seeks to provide a theoretical foundation and translational direction for the development of precision therapies targeting rheumatic autoimmune disorders.

## 2. T_RM_ Cell Biology in Rheumatoid Immune Diseases

### 2.1. Development and Maintenance Mechanisms of T_RM_ Cells

#### 2.1.1. Key Regulatory Signals

T_RM_ cell differentiation is influenced by both intrinsic transcriptional programs and extrinsic factors like local cytokines, adhesion molecules, and metabolic adaptations [24].

Local cytokines, as essential components of the tissue microenvironment, are crucial for both the differentiation and maintenance of T_RM_ cells [25]. Transforming growth factor-β (TGF-β) is a central inducer of the T_RM_ phenotype. Together with T cell receptor (TCR) signaling, TGF-β promotes the expression of integrin αE (CD103), thereby guiding CD8^+^ T cells toward a tissue-resident state [26]. Moreover, TGF-β fine-tunes downstream gene transcription through SMAD-dependent pathways, thereby ensuring that T_RM_ cells acquire and sustain long-term residency [27,28]. In coordination with other local cytokines, TGF-β suppresses the expression of circulation-associated genes, such as S1PR1 and KLF2, to further promote tissue retention [22]. Interleukin-15 (IL-15) is another indispensable cytokine for T_RM_ cell maintenance. Through activation of the JAK–STAT pathway, IL-15 upregulates anti-apoptotic molecules including Bcl-2 and enhances metabolic activity, thereby supporting T_RM_ cell survival, proliferation, and effector function [29]. IL-15 also sustains mitochondrial fatty acid oxidation (FAO), enabling T_RM_ cells to preserve energy homeostasis and resist apoptosis over prolonged periods [30,31]. Additionally, in certain tissues, interleukin-7 (IL-7) signals through CD127 and downstream JAK–STAT5 to further promote T_RM_ cell longevity and homeostasis [30,31]. Additionally, in certain tissues, IL-7 engages its receptor CD127 to activate downstream JAK–STAT5 signaling, further promoting T_RM_ cells longevity and homeostasis [32,33]. Collectively, TGF-β, IL-15, and IL-7 reshape transcriptional and signaling networks by repressing egress-associated genes while activating survival and metabolic programs, ultimately orchestrating T_RM_ cell differentiation, tissue residency, and persistence [25].

Another crucial determinant of long-term tissue residency in T_RM_ cells is their high expression of adhesion molecules [29]. Among these, CD103 is the most characteristic, mediating physical anchoring of T_RM_ cells to epithelial E-cadherin [34,35]. TGF-β signaling induces CD103 expression in CD8^+^ T cells, conferring the hallmark tissue-resident phenotype [28]. In addition, CD103 can interact with local extracellular matrix (ECM) components, such as collagen, to further stabilize T_RM_ cell retention in collagen-rich tissues [34]. Collagen-binding receptors also contribute indispensably to T_RM_ cell positioning and stability, helping cells maintain spatial organization and interact dynamically with their microenvironment [9]. These adhesion molecules collectively form a structural framework that enables rapid responses to local antigen re-exposure while sustaining long-term immune surveillance [13]. Moreover, adhesion molecules participate in intracellular signaling cascades that modulate metabolism and effector function, establishing positive feedback loops that reinforce T_RM_ cell differentiation and persistence [36].

#### 2.1.2. Metabolic Adaptation Mechanisms

To persist long-term in tissues, T_RM_ cells must adapt to environmental challenges such as nutrient scarcity, hypoxia, and metabolic stress. They achieve this through specialized metabolic adaptation mechanisms that optimize energy utilization and survival [37]. Among these, FAO plays a central role. Multiple studies have shown that under nutrient-deprived conditions where amino acids and glucose are limited, T_RM_ cells rely on Carnitine Palmitoyltransferase 1A (CPT1a)-mediated transport of fatty acids into mitochondria for β-oxidation [24,38]. Since T_RM_ cells often inhabit nutrient-poor tissue niches, lipid metabolism becomes a key energy source to sustain their long-term functionality [39].

Hypoxia is another prominent feature of tissue environments, profoundly influencing T_RM_ cell development and maintenance [40]. Under low oxygen tension, hypoxia-inducible factor-1α (HIF-1α) stabilizes and becomes transcriptionally active, initiating adaptive gene expression programs [41]. HIF-1α not only upregulates glycolytic enzymes but also modulates FAO and mitochondrial function, coordinating metabolic reprogramming and energy homeostasis within T_RM_ cells [36,42,43]. By increasing glucose uptake and lactate production, HIF-1α helps partially compensate for hypoxia-induced energy deficits, while its cooperative regulation with FAO supports T_RM_ cell survival in oxygen- and nutrient-limited environments [41]. Moreover, HIF-1α regulates ECM remodeling and adhesion molecule expression, providing structural and metabolic support that enables T_RM_ cells to remain stably resident under hypoxic conditions [25,44]. During this adaptive process, T_RM_ cells must simultaneously manage hypoxia-induced metabolic stress and integrate regulatory signals from local cytokines and adhesion molecules. Consequently, T_RM_ cells establish a highly integrated regulatory network that coordinates metabolism, signaling, and survival, ensuring stable residency and robust immune responsiveness within tissues [45].

The development and maintenance of T_RM_ cells depend on the synergistic regulation of local cytokines, adhesion molecules, and metabolic adaptation mechanisms, which together form a tightly interconnected immunoregulatory network [28,46]. Specifically, TGF-β functions as a central regulator—inducing CD103 expression to strengthen epithelial adhesion while reshaping cellular metabolism by activating FAO, thus enabling T_RM_ cells to sustain energy homeostasis and survival in hypoxic, nutrient-deprived environments [25,47]. IL-15 and IL-7 complement this process by activating anti-apoptotic pathways such as PI3K–AKT, further extending T_RM_ cell lifespan [48]. Meanwhile, adhesion molecules such as integrin αEβ7 not only provide structural anchorage but also mediate mechanotransducive signaling that coordinates the expression of metabolic genes, thereby promoting metabolic reprogramming [49,50]. Through these multidimensional interactions, T_RM_ cells acquire unique tissue-residency properties that enable them to function as a potent immune barrier within epithelial and barrier tissues, mounting rapid and durable responses against local pathogen invasion or tumor dissemination and achieving precise tissue-specific immune protection [51].

### 2.2. Heterogeneity and Disease Specificity of T_RM_ Cells

#### 2.2.1. Tissue-Distribution Heterogeneity

T_RM_ cells display marked differences in phenotypic markers, transcriptional programs, and functions across organs. For example, T_RM_ cells in the skin, lung, and intestine commonly express CD69, whereas CD103 expression varies substantially: CD8^+^ T_RM_ cells in the skin and intestinal epithelium often express high levels of CD103, while lung T_RM_ cell populations are predominantly CD103-negative or low-expressing [52,53]. This heterogeneity reflects tissue-specific shaping of cell fate by local microenvironments; cell–cell interactions, cytokine milieus, and extracellular-matrix composition collectively drive tissue-tailored transcriptional maturation and phenotypes [37,54].

Even within the same organ, distinct T_RM_ cell subpopulations are evident. In the skin, for instance, T_RM_ cells can be subdivided into epidermal and dermal compartments: epidermal T_RM_ cells typically exhibit high CD103 expression and stronger cytotoxic potential, whereas dermal T_RM_ cells tend to display more active cytokine production and proliferative capacity [52,55]. In addition, T_RM_ cells in the lung can also be divided into subpopulations that are retained stably in the lung interstitium and are continuously replenished by circulating T cells, which play different roles in antigen reencounter and continuous immune surveillance [9,56].

#### 2.2.2. Functional Heterogeneity

Beyond spatial diversity, T_RM_ cells also differ functionally. One subset is characterized by robust production of IFN-γ and TNF-α and exhibits potent pro-inflammatory and cytotoxic activities [57,58]; Another subset expresses immunoregulatory molecules such as IL-10 and Programmed death receptor 1 (PD-1) and contributes to local protection and immune modulation [59,60,61,62]. In addition, upon antigenic stimulation, a fraction of T_RM_ cells can exit their tissue of residence and enter the circulation as so-called “ex- T_RM_ cells,” which may display heightened proliferative capacity and multipotency [63,64].

##### Pro-Inflammatory T_RM_ Cells: Drivers of Tissue Damage

Pro-inflammatory T_RM_ cells serve as frontline responders in tissues. Upon antigenic or inflammatory cues, these cells rapidly secrete high levels of IFN-γ and TNF-α, promoting recruitment/activation of effector leukocytes and triggering inflammatory cell death and matrix degradation in resident cells, thereby exacerbating pathology [20,65,66]. In barrier sites such as skin, gut, and liver, abundant CD8^+^CD69^+^CD103^+^ T_RM_ cells are positioned within the epithelium or mucosal basal zones, poised to respond to reinfection or tissue injury and to promptly release IFN-γ and TNF-α [16,21]. Following influenza infection, a CD49a^+^CD103^+^CD8^+^ T_RM_ cell subset in the lung produces higher levels of IFN-γ than other T cells, contributing to viral clearance but also to local epithelial damage [67]. In psoriatic arthritis, synovial fluid is enriched for CD69^+^CD103^+^CD8^+^ T_RM_ cells that secrete IFN-γ and TNF-α and, together with IL-17A, induce cartilage-degrading enzymes, thereby directly promoting cartilage destruction [68].

##### Regulatory T_RM_ Cells: Guardians of Local Protection

T_RM_ cells not only exert potent pro-inflammatory effects but can also adopt regulatory or inhibitory phenotypes that limit excessive inflammation and promote tissue repair. Regulatory T_RM_ cells express IL-10, PD-1, and other inhibitory molecules, playing a key role in maintaining tissue balance and preventing immune-related damage. IL-10–producing T_RM_ cells rapidly suppress overactivation of macrophages and other effector cells, thereby mitigating tissue injury [69]. PD-1/PD-L1 signaling negatively regulates T_RM_ cell activity, curbing excessive IFN-γ and TNF-α production and facilitating tissue repair [21]. In secondary influenza challenge models, PD-1^+^ T_RM_ cells help balance antiviral immunity with pulmonary immunopathology; blockade of PD-1 causes over-expansion of T_RM_ cells and worsened tissue injury [70]. In a non-human primate model of SARS-CoV-2 infection, IL-10 signaling both promotes T_RM_ cell formation and limits excessive T-cell expansion, thereby balancing antiviral responses with tissue protection [60]. In the pancreas, T_RM_ cells maintain local immune homeostasis via the PD-1/PD-L1 axis, preventing autoimmune pancreatitis [71]. IL-10^+^ T_RM_ cells limit T-cell–mediated damage in cutaneous wound and intestinal inflammation models, promoting wound healing and mucosal repair [72]. Thus, therapeutically enhancing IL-10^+^/PD-1^+^ T_RM_ cells—or mimicking their functions—may offer new strategies for autoimmune diseases such as psoriatic arthritis and rheumatoid arthritis.

### 2.3. Disease-Specific T_RM_ Cells Revealed by Single-Cell Sequencing

Recent single-cell and histological studies have identified disease-specific T_RM_ cell subsets across rheumatic autoimmune conditions, showing that distinct chemokine-receptor repertoires and spatial localization underpin disease-driving programs.

In RA, single-cell sequencing of synovial tissue has revealed a phenotypically unique subset of PD-1^+^GPR56^+^ tissue-resident T cells co-expressing classical T_RM_ cell markers such as CXCR6, CD69, and LAG3. These cells show significant clonal expansion and are enriched in ACPA-positive patients, indicating antigen-driven persistence [73]. Earlier work indicated that synovial CXCR6^+^ T_RM_ cells can persist long-term within the joint microenvironment, continuously producing pro-inflammatory and regulatory mediators under inflammatory conditions, thereby sustaining chemokine/cytokine networks and, via crosstalk with B cells and other leukocytes, promoting chronicity and relapse of synovitis [74]. Integrated single-cell transcriptomics and TCR sequencing further indicate functional plasticity: these T_RM_ cells support B-cell autoantibody production and can also contribute directly to cartilage and bone damage via cytotoxic pathways and amplification of local inflammation [75]. These insights nominate CXCR6-mediated retention or effector programs as potential precision-targeting strategies to control RA synovitis and joint destruction.

In SLE, single-cell analyses of skin lesions have identified a population of T_RM_ cells expressing the skin-homing receptor CCR10. These CCR10^+^ T_RM_ cells localize to the epidermis and peri-follicular niches, where CCR10-dependent homing and retention enable long-term persistence and production of IFN-γ, IL-17, and other inflammatory mediators that exacerbate chronic cutaneous inflammation [76]. Single-cell data suggest interactions between CCR10+ T_RM_ cells and dendritic or B cells. These interactions may enhance antigen presentation, sustain T-cell activation, and contribute to autoimmunity [77]. While no clinical trials directly targeting CCR10^+^ T_RM_ cells have yet been registered, current single-cell evidence supports their importance in SLE skin pathology and points to CCR10-mediated skin-homing pathways as candidate therapeutic targets.

SSc is a systemic autoimmune disease characterized by fibrosis of the skin and internal organs [78]. While no clinical trials directly targeting CCR10^+^ T_RM_ cells have yet been registered, current single-cell evidence supports their importance in SLE skin pathology and points to CCR10-mediated skin-homing pathways as candidate therapeutic targets [79]. More recent large-scale single-cell work confirmed immune-cell compositional abnormalities and identified immune subsets associated with clinical heterogeneity, including T cells with resident/homing phenotypes [80]. In SSc-associated interstitial lung disease (SSc-ILD), studies further report enrichment of a CD8^+^ T-cell subset with type-II interferon features in both blood and lung tissues, whose migratory behavior may contribute to ILD progression [81]. These findings imply that T_RM_-like populations may participate in local immune dysregulation within SSc target organs.

In pSS, multiple single-cell sequencing studies have shown that the distribution and functional status of immune cells in the salivary glands and other exocrine glands play a key role in the destruction of glandular structure and loss of secretory function. Notably, CD8^+^ T_RM_ cells are markedly expanded in salivary glands, expressing canonical residency markers (CD69, CD103) along with high activation signatures (e.g., HLA-DR) and cytotoxic mediators (e.g., IFN-γ, granzyme B), thereby damaging glandular epithelium through direct cytotoxicity and pro-inflammatory cytokines [82,83]. These locally expanded T_RM_ cells also interact closely with dendritic cells and B cells, amplifying inflammatory cytokine and chemokine networks and culminating in glandular dysfunction and reduced secretion [82]. Single-cell transcriptomics further show that T_RM_ cells can persist locally without continuous peripheral input; their sustained inflammatory state promotes epithelial apoptosis/dysfunction, clinically manifesting as xerostomia and xerophthalmia [83].

In summary, single-cell technology reveals the disease-specificity and pathological role of T_RM_ cells in different autoimmune diseases. In RA, synovial CXCR6^+^ T_RM_ cells sustain local inflammation and drive joint damage via chemokine and cytokine outputs; in cutaneous SLE, CCR10-high T_RM_ cells persist in skin and exacerbate local responses through inflammatory mediators; in SSc, perivascular T_RM_ cells may contribute to fibroblast activation and fibrosis (e.g., via TGF-β–linked pathways); and in pSS, glandular CD8^+^ T_RM_ cells inflict epithelial injury through cytotoxic and inflammatory mechanisms, leading to secretory failure. Integrating single-cell with spatial omics will further clarify these mechanisms and accelerate the development of T_RM_ cell-targeted immunotherapies tailored to disease context (As shown in Table 1).

## 3. T_RM_ Cell-Driven Pathogenic Mechanisms

In recent years, numerous studies have shown that T_RM_ cells not only protect the host by preventing pathogen re-encounter but also play pivotal roles in the pathogenesis of autoimmune diseases. In disorders such as RA [84,85] and SLE [86], T_RM_ cells within tissues secrete inflammatory mediators, cytokines, and chemokines that drive uncontrolled local inflammation, culminating in tissue destruction, fibrosis, and organ dysfunction (As shown in Figure 1).

### 3.1. T_RM_ Cell-Mediated Pathogenic Mechanisms in Rheumatoid Arthritis (RA)

#### 3.1.1. Synovial T_RM_ Cells–Fibroblast Crosstalk and Promotion of Osteoclastogenesis

Within RA synovium, T_RM_ cells are locally enriched in inflamed joints and display distinct residency phenotypes; these cells engage in close cell–cell interactions with synovial fibroblasts [87]. Synovial T_RM_ cells exhibit pronounced clonal expansion and, upon recognition of autoantigens in situ, rapidly secrete multiple pro-inflammatory cytokines, including GM-CSF [88]. GM-CSF is a key inflammatory mediator that stimulates synovial fibroblast proliferation and activation and directly acts on macrophages and osteoclast precursors to induce their differentiation into mature osteoclasts, thereby promoting bone erosion within the joint [88,89]. Thus, immune–stromal interactions driven by T_RM_ cell-derived inflammatory factors are central to RA joint damage. In addition, T_RM_ cell-derived IL-17 and other cytokines form a positive feedback loop with synovial fibroblasts, which, in turn, produce IL-6, CXCL12, and RANKL. As a critical signal for osteoclast differentiation, excessive RANKL expression directly accelerates bone resorption and joint destruction [75,90]. This cytokine cascade amplifies into a self-reinforcing pathological circuit, sustaining inflammation and ultimately leading to irreversible tissue injury [90].

#### 3.1.2. Role of the IL-23/IL-17 Positive Feedback Loop in Maintaining Chronic Inflammation

Beyond GM-CSF–mediated osteoclast activation, T_RM_ cells and local antigen-presenting cells (APCs) establish a positive feedback circuit centered on IL-23 and IL-17, which maintains and exacerbates synovial inflammation [87,91]. In this loop, dendritic cells and macrophages secrete IL-23, which effectively induces T_RM_ cells and Th17 cells to produce IL-17. IL-17 not only directly activates synovial fibroblasts to release additional inflammatory mediators and RANKL but also promotes local GM-CSF production, thereby closing the inflammatory loop within the joint microenvironment [92,93]. Consequently, even when systemic inflammatory mediators decline, the intra-articular cytokine network can self-sustain a highly inflamed state, driving chronicity and progressive tissue destruction [20].

#### 3.1.3. Extra-Articular Manifestations

The role of pulmonary T_RM_ cells in RA-associated interstitial lung disease RA is not only limited to joint inflammation, but its extra-articular manifestations have also attracted much attention, among which RA-associated interstitial lung disease (RA-ILD) is one of the important manifestations of significant pathogenicity [94]. Emerging evidence suggests that pulmonary T_RM_ cells also mediate local immune responses and may act as “ignition switches” in fibrotic lesions. Upon environmental or autoantigenic stimulation, lung T_RM_ cells are activated to secrete inflammatory factors such as IL-17 and GM-CSF, which induce profibrotic mediator release from interstitial cells and activate fibroblasts [95,96]. This local immune activation partially mirrors intra-articular mechanisms, implying that pulmonary T_RM_ cells may promote interstitial inflammation and fibrosis through cytokine networks analogous to those in synovium, ultimately impairing lung function [95]. Targeting T_RM_ cells and their associated cytokine networks could, therefore, provide new therapeutic entry points for RA and its extra-articular complications [94].

### 3.2. T_RM_ Cell Pathogenesis in Systemic Lupus Erythematosus (SLE)

#### 3.2.1. Long-Term Retention of Skin T_RM_ Cells and Production of Type I Interferon

In SLE—especially cutaneous manifestations such as cutaneous lupus erythematosus and discoid lupus—T_RM_ cells have garnered increasing attention. Cutaneous T_RM_ cells, characterized by CD69 and CD103 expression, exhibit long-term residency that supports sustained local immune activation [86,97]. Following triggers such as ultraviolet irradiation, these persistent T_RM_ cells can be reactivated to produce large amounts of type I interferons (e.g., IFN-α). Type I interferons enhance antigen presentation and inflammation and activate neighboring keratinocytes to produce additional inflammatory mediators, thereby establishing a self-sustaining pro-inflammatory network in the skin [98]. Consequently, persistent activation of cutaneous T_RM_ cells is a key driver of chronic inflammation, lesion persistence, and frequent relapse in SLE skin disease.

#### 3.2.2. The Formation of T_RM_ Cells and B Cell Immune Synapses in the Kidney Promotes the Production of Autoantibodies

Beyond the skin, lupus nephritis represents one of the most severe SLE manifestations. Recent studies indicate that renal T_RM_ cells participate in local immune regulation and directly promote autoantibody production through interactions with B cells [99]. In the kidney, T_RM_ cells can form immunological synapses with B cells via CD40/CD40L interactions, delivering potent co-stimulatory signals that drive B-cell proliferation and differentiation into antibody-secreting plasma cells [100,101]. Concurrently, T_RM_ cell-derived IL-21 further activates B-cell secretory pathways, enhancing immunoglobulin production and class switching, thereby exacerbating pathogenic autoantibody generation [100]. This T_RM_ cell–B-cell crosstalk is a critical component of SLE immunopathology, fostering immune-complex deposition and correlating with the severity of organ damage (e.g., glomerular injury).

### 3.3. T_RM_ Cell-Mediated Pathogenic Mechanisms in Systemic Sclerosis (SSc)

#### 3.3.1. T_RM_ Cell-Derived TGF-β as a Driver of Cutaneous and Vascular Fibrosis

SSc is clinically defined by widespread fibrosis, presenting as skin thickening, internal-organ fibrosis, and vasculopathy. Recent studies suggest that aberrantly activated T_RM_ cells may directly contribute to fibrogenesis in SSc through secretion of profibrotic cytokines such as TGF-β [102]. TGF-β is a central mediator of fibrosis, promoting fibroblast proliferation and differentiation into myofibroblasts and inducing extracellular matrix deposition (e.g., collagen), which culminates in tissue stiffening and functional impairment [103]. Although direct human evidence that T_RM_ cells produce TGF-β in SSc remains limited, the multifaceted secretory capacity of T_RM_ cells in other autoimmune settings, together with the highly inflamed SSc lesions, supports the hypothesis that aberrant T_RM_ cell activation may constitute an important local source of TGF-β that accelerates fibrosis [79].

#### 3.3.2. Epigenetic Reprogramming Underlies Aberrant T_RM_ Cell Activation

In addition to profibrotic cytokine production, epigenetic dysregulation is thought to contribute to abnormal T_RM_ cell function in SSc. T cells from patients with SSc exhibit pronounced DNA-methylation abnormalities and histone-modification defects, leading to dysregulated expression of immune regulatory genes and a persistently activated T_RM_ cell-like state [104,105]. Such epigenetic reprogramming disrupts immune tolerance and predisposes T_RM_ cells to produce pro-inflammatory and profibrotic cytokines—including TGF-β—thereby aggravating local fibrosis and endothelial pathology [106]. Thus, in SSc, aberrant T_RM_ cell activation appears to be co-driven by exogenous inflammatory cues and intrinsic epigenetic factors, forming a self-sustaining fibrotic circuit that severely compromises the function of skin and internal organs [107,108].

### 3.4. T_RM_ Cell-Mediated Pathogenic Mechanisms in Primary Sjögren’s Syndrome (pSS)

In pSS, pathological changes primarily affect salivary and lacrimal glands, leading to acinar destruction and loss of secretory function, clinically manifesting as xerostomia and xerophthalmia. Recent studies highlight T_RM_ cells residing in salivary glands as key mediators of this process [109]. These glandular T_RM_ cells persist long-term and, upon activation, can secrete lymphotoxin-α (LTα)—a TNF-family cytokine with direct tissue-damaging potential [110]. LTα establishes an early pro-inflammatory milieu within salivary glands; even in the absence of overt lymphocytic infiltration, glandular secretion can be impaired due to LTα-mediated immune injury [111,112]. As disease progresses, sustained T_RM_ cell activity disrupts glandular architecture, damaging acini and ducts and ultimately causing irreversible functional loss, consistent with clinical xerostomia [83,113].

## 4. Therapeutic Targeting of T_RM_ Cells in Rheumatoid Immune Diseases

### 4.1. Drug Repurposing

#### 4.1.1. JAK Inhibitors

The survival and proliferation of T_RM_ cells depend substantially on IL-15 signaling, which primarily activates downstream effectors via the JAK–STAT pathway—especially STAT5—to support T_RM_ cell maintenance and function [114]. Current studies indicate that JAK inhibitors effectively block IL-15 signal transduction, thereby reducing T_RM_ cell survival and the secretion of pro-inflammatory cytokines, with notable therapeutic benefits in inflammatory skin diseases such as vitiligo and psoriasis [115]. In addition, JAK inhibition attenuates antigen-triggered local inflammatory responses, providing a preliminary basis for subsequent combination strategies [116,117].

#### 4.1.2. Anti-IL-17/23 Antibodies

IL-17 and IL-23 are pivotal in regulating T_RM_ cell functions and sustaining an inflammatory microenvironment [118]. In psoriasis, tissue-resident T_RM_ cells represent a key source of IL-17, whereas IL-23 provides essential survival and functional support; the IL-23/IL-17 axis relies on a local “inflammatory memory pool” formed by T_RM_ cells to drive relapse [119]. Clinical studies have shown that anti-IL-23 antibodies (e.g., risankizumab) and anti-IL-17 antibodies (e.g., secukinumab) achieve significant efficacy in psoriasis and other Th17-mediated diseases. Their durable remission partly reflects indirect modulation of T_RM_ cells, interrupting cytokine feedback circuits and dampening T_RM_ cell-driven pro-inflammatory functions [120,121].

#### 4.1.3. Low-Dose Radiotherapy (LDRT)

As a local modality, LDRT shows potential for the selective reduction of T_RM_ cells. Traditionally used for anti-inflammatory and anti-proliferative effects, recent evidence suggests that precise dose control may locally “debulk” or decrease radio-tolerant T_RM_ cell populations while sparing other immune cells, thereby alleviating local inflammation and improving clinical status [122]. This approach is particularly applicable to focal inflammatory conditions, such as RA [123], and psoriasis [122], and may complement systemic therapies.

### 4.2. Emerging Targeted Strategies

Although repurposed agents can partially restrain T_RM_ cell-mediated inflammation, complete eradication remains challenging due to the tissue-retentive nature of T_RM_ cells. Consequently, new strategies are being developed to disrupt residency programs and to intervene in key metabolic pathways (As shown in Table 2).

#### 4.2.1. Disrupting Residency Signals

T_RM_ cell localization partly depends on responsiveness to sphingosine-1-phosphate (S1P). S1PR1 is critical for regulating lymphocyte egress and tissue trafficking. Pharmacologic activation of S1PR1 (e.g., fingolimod) can promote T_RM_ cells egress from niches, reduce the density of local inflammatory cells, and ameliorate focal immune inflammation. By altering intra-tissue distribution, this strategy offers a new point of intervention that can be combined with other immunomodulators [55]. Although most data derive from inflammatory bowel disease and related contexts, the underlying mechanisms extend to other rheumatic diseases characterized by recurrent focal inflammation [131].

Integrin αEβ7 (linked to CD103 expression) is indispensable for T_RM_ cell adhesion to epithelial cells. Inhibiting this integrin can disrupt T_RM_ cell–tissue interactions, facilitating egress or reducing residency stability and thereby diminishing chronic local inflammation [132]. Preclinical data suggest that integrin inhibitors may become useful tools against inflammatory skin or rheumatic diseases [114]. In SSc, TGF-β is a key fibrogenic cytokine, and multiple αv integrins (e.g., αvβ1, αvβ3, αvβ5, αvβ6, αvβ8) participate in its activation. Thus, targeting αv-integrin–mediated latent TGF-β activation has emerged as a more prominent antifibrotic strategy than αEβ7 blockade and represents a promising approach in SSc [133].

#### 4.2.2. Metabolic Interventions

T_RM_ cells exhibit heightened energy demands to sustain long-term survival and effector competence, with a particular reliance on fatty-acid oxidation (FAO). In gastric adenocarcinoma, CD8^+^CD103^+^ T_RM_ cells undergo metabolic reprogramming and depend on FAO, with CPT1a acting as a rate-limiting enzyme; rewiring metabolism prolongs T_RM_ cell lifespan and enhances anti-tumor immunity [134]. Similarly, CPT1a-mediated FAO is critical in other tissue-resident immune populations. A 2024 study in lupus nephritis models reported reduced CPT1a expression in renal and circulating macrophages, impairing efferocytosis and exacerbating renal inflammation [124]. These data suggest that immune cell–specific metabolic targeting (e.g., modulating CPT1a) offers a promising avenue in autoimmunity. By analogy, targeting CPT1a in T_RM_ cells may provide a metabolic handle for intervention and could be combined with other immunoregulatory therapies.

Metformin, a widely used antihyperglycemic agent, activates AMPK and inhibits mTORC1, downregulating anabolic and proliferative programs and potentially affecting the survival and function of metabolically dependent immune subsets [135]. Although direct evidence for T_RM_ cell-specific effects is lacking, metformin’s immunomodulatory impact has attracted attention. Preclinical studies indicate modulation of T-cell subset balance (e.g., Th17/Treg) in multiple rheumatic models [125], and clinical data in SLE show reduced relapse risk and steroid-sparing trends with metformin add-on therapy, alongside improvements in patient-reported outcomes [126]. These findings support the hypothesis that systemic immune-metabolic tuning may indirectly influence T_RM_ cell-involved chronic inflammation and justify further exploration in rheumatic disease.

### 4.3. Microenvironment-Remodeling Strategies

Long-term T_RM_ cell persistence relies not only on intrinsic survival programs but also on extrinsic support from the local microenvironment—including ECM, chemokines, and neighboring immune cells. Remodeling this niche to weaken trophic support for T_RM_ cells represents an innovative and promising therapeutic direction.

#### 4.3.1. Smart Nanoparticle Delivery of siRNA

This approach employs functionalized nanocarriers to deliver small interfering RNAs (siRNAs) to targeted cells within lesions, silencing pathogenic genes to perturb T_RM_ cell survival and function. Multiple smarts, stimulus-responsive nanoparticles have been developed. For example, macrophage-targeted polymeric polymersomes can sequentially deliver TNF-α siRNA (siTNFα) followed by dexamethasone. In the collagen-induced arthritis (CIA) model, this “temporal nanotherapy” reverses glucocorticoid resistance via TNF-α silencing and then achieves synergistic anti-inflammatory and joint-protective effects with co-administration [127]. Another microenvironment-driven, deformable self-assembling nanoplatform responds to elevated MMP-2 and acidic pH in RA joints. Upon arrival, the platform deforms to form a physical barrier limiting inflammatory cell infiltration and precisely releases payloads (e.g., triptolide, metformin), thereby remodeling the vascular–immune–inflammatory milieu. Preclinical studies demonstrate specificity and safety of nanoparticle-mediated siRNA delivery for local gene modulation, supporting future clinical translation [128].

#### 4.3.2. Synthetic-Biology Engineering of Fibroblasts

An additional strategy seeks to reprogram local fibroblasts using synthetic-biology tools to create in situ “drug factories” that continuously secrete therapeutic proteins. In rheumatic diseases—particularly RA—engineering fibroblasts to reconfigure the joint microenvironment is an active frontier [129]. In RA, aberrant activation of fibroblast-like synoviocytes (FLS) is central to synovial inflammation and bone destruction [130]; synthetic-biology interventions aim for precise control of these cells. In the future, such local microenvironmental control can be combined with systemic therapies (small molecules or biologics) to achieve complementary mechanisms and synergistic efficacy, offering a more comprehensive therapeutic paradigm for rheumatic autoimmune diseases [136].

## 5. Challenges and Future Perspectives

T_RM_ cells, a subset of T cells that permanently reside in barrier tissues and non-lymphoid organs, play key roles in maintaining chronic inflammation and driving relapse in rheumatic autoimmune diseases such as RA, psoriatic arthritis, and SLE. Although targeting T_RM_ cells offers new promise for durable disease control, clinical translation faces multiple challenges and will depend on advances in technology and personalized treatment strategies.

### 5.1. Challenges for Clinical Translation

#### 5.1.1. Difficulties in Tissue-Specific Targeting

Marked heterogeneity of T_RM_ cells across tissues complicates precise targeting. Studies indicate substantial functional differences across organs; for example, skin T_RM_ cells may depend on antigen-specific deletion to sustain protective immunity, whereas synovial T_RM_ cells appear less dependent on this mechanism. This raises the prospect of tissue-differentiated interventions but also underscores the difficulty of selectively modulating T_RM_ cells across distinct tissues [55,137]. In inflammatory arthritis, synovial T_RM_ cells can persist in remission and establish “joint-specific memory,” thereby mediating disease flares. Given that indiscriminate T_RM_ cell depletion may compromise local immune homeostasis—and that T_RM_ cell heterogeneity spans functions and regulatory programs—no broadly applicable strategy yet balances effective suppression of pathogenic inflammation with preservation of protective immunity. This constitutes a central complexity in current clinical translation [137].

#### 5.1.2. Potential Impacts of T_RM_ Cell Depletion on Immune Memory

Potential erosion of immune memory is a key concern in therapeutic T_RM_ cell depletion. As sentinels of barrier immunity, T_RM_ cells form the first defensive layer against local pathogen entry. Non-selective depletion may weaken local defenses and increase opportunistic infections [138]. Even putatively selective strategies risk diminished anti-infective memory and suboptimal vaccine responses if specificity is insufficient, leading to heightened infection risk and reduced vaccine efficacy [137]. In tissues such as joints and skin, T_RM_ cells combine pro-inflammatory functions with immune surveillance; thus, intervention requires careful balancing of inflammatory control with baseline protection [137]. Achieving precise control of inflammation-related T_RM_ cells while maintaining local and systemic immune homeostasis has therefore become a central translational question.

### 5.2. Key Technological Directions

#### 5.2.1. Spatial Transcriptomics: Mapping T_RM_ Cells–Stroma Interactions

Spatial transcriptomics has emerged as a powerful tool to resolve cellular positioning and interactions in situ, enabling precise mapping of T_RM_ cell niches and their crosstalk with neighboring cells. In lupus nephritis (LN), spatial profiling revealed a distinct inflammatory–injury niche in the renal cortex enriched for VCAM1^+^ proximal tubule cells, myofibroblasts, and multiple immune lineages. VCAM1^+^ tubule cells actively recruit immune cells via chemokines (e.g., CXCL12) and signal with surrounding cells through TGF-β pathways, collectively shaping a pro-inflammatory, pro-fibrotic microenvironment [139]. In RA synovium, spatial analyses identified a key interaction between CD8^+^ T cells and endothelial cells via the CCL5–ACKR1 ligand–receptor axis, with spatial co-localization that facilitates transendothelial leukocyte trafficking and exacerbates synovitis [140].

#### 5.2.2. Organoid Models: Modeling Long-Term T_RM_ Cell Persistence and Optimizing Drug Discovery

Organoid systems recapitulate tissue architecture and function and are increasingly used to model immune–epithelial interfaces. A 2024 study established human intestinal immune organoids (IIOs) containing autologous T_RM_ cells. T_RM_ cells actively invaded and integrated into the epithelial barrier at an approximate 1:16 ratio—mirroring healthy gut distributions—and remained viable for ≥14 days under low-cytokine conditions [141]. In rheumatic disease research [142,143], patient-derived 3D organoids have reproduced fibroblast-like synoviocyte (FLS) hyperproliferation and cytokine cascades (e.g., TNF-α, IL-6). Single-cell sequencing confirms preservation of patient-specific transcriptional signatures, enabling mechanistic dissection and pharmacologic testing.

#### 5.2.3. AI-Guided Epigenetic Target Discovery for Precision Therapy

Artificial intelligence (AI) is increasingly applied to identify epigenetic targets. DNA methylation and other epigenetic marks play pivotal roles in autoimmune diseases, particularly RA and SLE. Pro-inflammatory cytokines and microenvironmental stress can reprogram methylation patterns, which may predict disease course, subtypes, and therapeutic response [144,145].

### 5.3. The Future of Individualized Therapy

#### 5.3.1. T_RM_ Cell Biomarker Assessment

Given their persistence and rapid activation in local inflammation, T_RM_ cell abundance and functional state correlate with disease activity. Assessment typically relies on canonical surface markers such as CD69 and CD103. In cutaneous lupus, lesional skin exhibits significantly increased CD69^+^ and/or CD103^+^ T_RM_ cells compared with healthy skin, directly indicating local accumulation in autoimmune lesions [97]. In psoriatic arthritis, a pro-inflammatory CD161 + CCR6^+^ type-17–like T_RM_ cell subset produces IL-17A, TNF-α, and IFN-γ, suggesting potential biomarker utility for distinguishing arthritis subtypes [68].

CD39^+^ T_RM_ cells represent a phenotypic subset whose expression levels reflect local inflammatory burden and dysregulated immune control [55]. In inflammatory bowel disease and RA, changes in CD39 expression correlate with clinical fluctuations, supporting CD39^+^ T_RM_ cell load as a candidate indicator of severity and prognosis [146]. High-resolution platforms—including flow cytometry, immunofluorescence, and spatial transcriptomics—now enable quantitative and phenotypic profiling of T_RM_ cell subsets.

#### 5.3.2. Combination Therapies

Onotherapy with immunosuppressants may be insufficient to fully eliminate or reprogram pathogenic T_RM_ cells. Therapies that selectively remove aberrant resident cells—such as engineered fusion antibodies, toxin conjugates, or cellular approaches—offer potential precision routes to target T_RM_ cells [55,147]. Recent work has used targeted lipid nanoparticles (tLNPs) to deliver mRNA and reprogram CD8^+^ T cells in samples from healthy donors and autoimmune patients, opening in vivo therapeutic avenues for cancer and autoimmunity [148]. However, cell removal alone rarely prevents relapse; microenvironmental repair is equally critical. Mesenchymal stem cell (MSC)-based immunomodulation and advanced biomaterials are gaining traction. In multiple sclerosis, MSC infusion induced T-cell anergy, reduced Th17 cells, and promoted antigen-specific Treg polarization, reversing Th17/Treg imbalance, dampening inflammation and demyelination, and restoring tolerance [149].

Emerging biomaterials—such as smart nanoparticles and instructive scaffolds—enable localized, sustained delivery of anti-inflammatory agents, promote matrix repair, and support tissue regeneration. A 2025 study engineered an immunomodulatory nanomedicine (MP@NEs/CT) co-delivering multi-epitope citrullinated autoantigens and triptolide (TPL), achieving effective RA control and inducing antigen-specific tolerance [150].

Although the direct “clearance + repair” combination strategy is still in the exploration stage in the field of T_RM_ cells and rheumatology, some cutting-edge studies have shown a trend of synergistic integration of therapies with different mechanisms of action, which provides ideas for real combined strategies in the future. The 2025 study mentioned above is a good example. The nanodrug (MP@NEs/CT) designed in this study carries two functions at the same time: on the one hand, it carries a low-dose triptolidine (TPL) that can eliminate abnormally activated immune cells (such as Th1 and Th17 cells) through immunosuppression; On the other hand, the multiepitope citrullinated autoantigen it carries can induce antigen-specific immune tolerance and promote the proliferation of regulatory T cells (Treg) and B cells (Breg) with protective effects, thereby restoring immune balance [150]. This idea of integrating “clearance” and “regulation” functions in the same delivery system is very close to the synergistic concept of “removing disease-causing cells” and “repairing the microenvironment”.

Current research on rheumatic immune diseases is rapidly moving towards precise and individualized treatment. In the key direction of technological innovation, spatial transcriptomics technology has provided a powerful tool for elucidating the spatial distribution and interaction between T_RM_ cells and stromal cells in the inflammatory region, and laid a foundation for revealing the cellular communication network in the lesion microenvironment. By reproducing the complex local microenvironment in vivo, organoid models can not only simulate the mechanism of long-term survival of T_RM_ cells, but also provide a more reliable in vitro testing platform for new drug screening and efficacy optimization. At the same time, in the future of personalized therapy, through the fine evaluation of biomarkers such as CD39^+^ T_RM_ cells, it can directly reflect the severity of the disease and the inflammatory burden, so as to achieve dynamic monitoring and treatment plan adjustment. In the future, the deep integration of multiple technologies and clinical translational research will further promote the leap from basic mechanism research to precision treatment strategies, ultimately enabling doctors to formulate personalized and efficient treatment plans based on the unique immune microenvironment, epigenetic characteristics and cell distribution status of each rheumatic immune disease patient, thereby reducing the risk of recurrence and improving the quality of life of patients.

## 6. Conclusions

T_RM_ cells have firmly established themselves as central drivers of chronicity and relapse in rheumatic autoimmune diseases. This review has synthesized evidence illustrating how these long-lived tissue guardians, crucial for peripheral immunity, become key effectors of pathology in diseases like RA, SLE, SSc, and pSS. Their unique biology—governed by cytokines (TGF-β, IL-15), adhesion molecules (CD103), and metabolic adaptations (fatty acid oxidation)—enables their long-term persistence in tissues, forming stable “lesional memories” that sustain inflammation even during systemic remission.

The pathogenic roles of T_RM_ cells are both diverse and disease-specific. Single-cell technologies have unveiled distinct subsets, such as synovial CXCR6^+^ T_RM_ cells in RA that fuel osteoclastogenesis and chronic synovitis via GM-CSF and the IL-23/IL-17 axis, and CCR10^+^ T_RM_ cells in SLE skin that act as reservoirs for type I interferon production. In addition, recent studies have shown that CD4^+^ T_RM_ cells, particularly TH1 and TH17 subsets, also contribute significantly to RA pathology by driving synovial inflammation, enhancing cytokine secretion (e.g., IFN-γ, IL-17), and promoting tissue damage through immune interactions with fibroblasts and endothelial cells. Their persistence in the joint microenvironment is regulated by factors such as TGF-β, IL-15, and metabolic pathways like fatty acid oxidation, similar to their CD8+ counterparts [151]. In pSS, cytotoxic CD8^+^ T_RM_ cells directly mediate glandular destruction, while in SSc, T_RM_ cells are implicated in promoting fibrosis, potentially through TGF-β secretion. This detailed understanding underscores that T_RM_ cells are not merely bystanders but active architects of tissue damage across different autoimmune contexts. Targeting T_RM_ cells presents a paradigm shift from broadly immunosuppressive strategies towards precision medicine. Near-term opportunities lie in drug repurposing: JAK inhibitors disrupt critical IL-15 survival signals, and anti-IL-17/IL-23 antibodies break pathogenic cytokine feedback loops. However, the future lies in novel mechanisms designed to undermine their residency. Strategies in development include disrupting tissue retention via S1PR1 agonists (e.g., fingolimod) or integrin inhibitors, and targeting their core metabolism through CPT1a inhibition. The most innovative approaches involve microenvironmental remodeling, such as smart nanoparticles for targeted siRNA delivery and synthetic biology to engineer reparative stromal cells, aiming to actively reset the diseased tissue niche rather than just suppress immunity.

While T_RM_-targeting strategies hold promise, challenges persist. The heterogeneity of T_RM_ cells across tissues makes it difficult to develop targeted therapies. A key concern is the potential impairment of protective immunity, as non-selective depletion of T_RM_ cells could compromise the defense against pathogens and weaken vaccine responses. Thus, the central challenge lies in selectively silencing pathogenic functions of T_RM_ cells while preserving their protective roles. Future progress will depend on the integration of cutting-edge technologies, including spatial transcriptomics to map T_RM_ cell interactions in lesions, patient-derived organoids for more relevant drug testing, and artificial intelligence to identify novel epigenetic and biomarker targets. The evaluation of T_RM_ cell-associated biomarkers, such as CD39, will be crucial for patient stratification and monitoring, facilitating personalized treatment strategies.

In perspective, T_RM_ cells represent a critical frontier in the quest for lasting disease control in rheumatic autoimmunity. While hurdles of specificity and safety exist, the converging advances in immunology and bioengineering are paving the way for a new generation of therapies. Targeting T_RM_ cells moves beyond transient immunosuppression, offering a transformative strategy to fundamentally reset local tissue immunity and bring the goal of long-term remission and potential cures within closer reach.

## Figures and Tables

**Figure 1 biomedicines-13-02945-f001:**
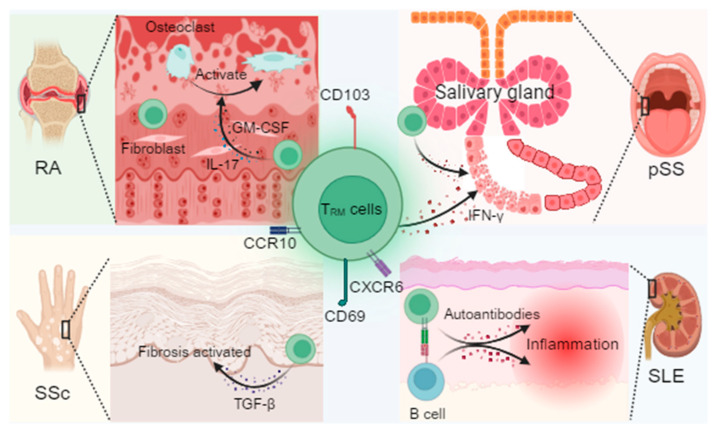
Mechanistic overview of T_RM_ cells and their pathogenic roles across autoimmune diseases. T_RM_ cell markers include CD69, CD103, CXCR6, and CCR10, maintained by cytokine signaling through transforming growth factor-β (TGF-β), interleukin-15 (IL-15), and interleukin-7 (IL-7). In rheumatoid arthritis (RA), synovial T_RM_ cells interact with fibroblasts, secrete granulocyte–macrophage colony-stimulating factor (GM-CSF), and promote osteoclast activation, contributing to joint erosion. In primary Sjögren’s syndrome (pSS), CD8^+^ T_RM_ cells in the salivary glands release cytotoxic mediators that damage glandular epithelium. In systemic sclerosis (SSc), T_RM_-derived TGF-β drives fibroblast activation and tissue fibrosis, accompanied by epigenetic reprogramming of local immune cells. In systemic lupus erythematosus (SLE), T_RM_ cells in the skin and kidney produce type I interferons and form immune synapses with B cells, facilitating autoantibody production. Together, these findings highlight T_RM_ cells as critical mediators of tissue inflammation, fibrotic remodeling, and disease-specific “inflammatory memory”. Created in BioRender. Min, D. (2025) https://BioRender.com/lxe8sih (accessed on 29 October 2025).

**Table 1 biomedicines-13-02945-t001:** Characteristics and functions of T_RM_ cells in different rheumatic immune diseases.

Disease	T_RM_ Cell Markers	Functional Subtypes	Pathogenic Mechanisms	References
Rheumatoid arthritis (RA)	CXCR6^+^	IFN-γ^+^/TNF-α^+^	GM-CSF secretion and IL-23/IL-17 axis activation enhance osteoclast activity	[73]
Systemic lupus erythematosus (SLE)	CCR10^+^	IFN-γ^+^/IL-10^+^	Continuous secretion of type I interferon activates B cells to promote autoantibody production	[76,77]
Systemic sclerosis (SSc)	Lung tissue: The memory phenotype is differentiated by CD45RO ↑ and the resting phenotype CD45RA ↑; (↑ indicates upregulation)	Lung tissue: CD8^+^ T_RM_ cells are the main focus;	Lung tissue: T_RM_/Treg-driven TCR signaling, T cell exhaustion, and epithelial interaction associate with fibrosis. Skin: T_RM_ cells-induced Th1/Th2 cytokine imbalance indicates chronic antigenic disruption of homeostasis.	[79,81]
Sjögren’s syndrome (pSS)	Skin: CD69, ITGAE, CCR7, SELL	Skin: CD4^+^/CD8^+^ T_RM_ cells and proliferative T_RM_ cell clusters	Disruption of the structure of the gland, leading to loss of secretory function	[82,83]

**Table 2 biomedicines-13-02945-t002:** Existing and potential T_RM_ cell-targeted therapy strategies.

Type	Medications/Methods	Mechanism	Applicable Diseases	Phase	References
Drug Repurposing	JAK inhibitors (Tofacitinib)	Block IL-15/JAK-STAT signaling → reduce T_RM_ cell survival and cytokine production	RA, SLE	Clinical application	[115,116,117]
	Anti-IL-17/23 antibodies (Secukinumab, Risankizumab)	Interrupt IL-23/IL-17 feedback loop → suppress T_RM_ cell-driven inflammation	RA, pSS	Clinical application	[120,121]
Residency-Targeting Approaches	S1PR1 agonist (Fingolimod)	Promote T_RM_ cell relocation and reduce local retention	IBD	Clinical research	[55]
Metabolic regulation	Integrin αEβ7 inhibitors (Etrolizumab)	Inhibition of T_RM_ cell fatty acid oxidation reduces survival	SLE	Preclinical research	[124]
	CPT1a inhibitors	Regulates T_RM_ cell metabolism through AMPK, limit _RM_ cells fatty-acid oxidation	RA, SLE	Clinical research	[125,126]
Microenvironment reshaping	Metformin	Targeted silencing of T_RM_ cell pro-inflammatory factors	RA	Animal Experiments	[127,128]
	Nanoparticles deliver siRNA	Induce anti-T_RM_ cell factor secretion	RA	Preclinical research	[129,130]

## Data Availability

No new data were created or analyzed in this study.

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
