# Peer review of "Tissue-Resident Memory T Cells in Rheumatoid Immune Diseases: Pathogenic Mechanisms and Therapeutic Strategies"

_biomedicines, 2025, doi:10.3390/biomedicines13122945_

Round 1
Reviewer 1 Report
Comments and Suggestions for Authors
The manuscript titled “Tissue-Resident Memory T Cells in Rheumatoid Immune Diseases: Pathogenic Mechanisms and Therapeutic Strategies” is comprehensive, well-researched, and scientifically up-to-date. It effectively integrates molecular and translational aspects, contributing to the understanding of TRM immunometabolism. However, minor revisions are necessary to meet the publication standards of the Biomedicine journal.
Comments:
- The abstract is lengthy and requires compression and refinement to enhance its focus and conciseness.
- The overall English quality is not good for publication-level writing. It’s more like a direct translation from Chinese, such as overly formal syntax and redundancy. Additionally, there are numerous typographical errors throughout the manuscript (especially in the abbreviation list). Some examples are as follows:
“Sstemic”
“Pmary”
“Sstemic lupus erythematosus”
“Sstemic sclerosis”
“Pmary Sjögren’s syndrome”
- Some abbreviations are missed, including GM-CSF, PD-1, CPT1a, etc.
- Ensure consistent style for cytokine and gene symbols (e.g., IL-17A, not IL17A)
- Figure 1, created using BioRender, appears to have a low resolution. Please provide a high-resolution version of Figure 1.
- Add a legend explaining abbreviations for Figure 1.
- Please separate the “References” column in Table 1-2.
- It is recommended to include a figure that provides an overview of therapeutic strategies, such as repurposed drugs and targeting metabolic and microenvironmental factors.
- Expand limitations and discuss the lack of in vivo validation for TRM-targeting strategies.
Author Response
Comments 1: The abstract is lengthy and requires compression and refinement to enhance its focus and conciseness.
Response 1: We thank the reviewer for their valuable feedback. We agree that the original abstract was overly lengthy and have thoroughly revised it to improve focus and clarity. Redundant wording has been removed, sentence structures have been streamlined, and overlapping descriptions have been consolidated to enhance conciseness without altering the scientific content. The revised abstract is now substantially more succinct while fully preserving all essential information and key messages. We believe these revisions have enhanced readability and ensure that the abstract is better aligned with the journal’s formatting standards.
Comments 2: The overall English quality is not good for publication-level writing. It’s more like a direct translation from Chinese, such as overly formal syntax and redundancy. Additionally, there are numerous typographical errors throughout the manuscript (especially in the abbreviation list). Some examples are as follows: “Sstemic” “Pmary” “Sstemic lupus erythematosus” “Sstemic sclerosis”“Pmary Sjögren’s syndrome”
Response 2: We sincerely thank the reviewer for pointing out the issues related to language quality and typographical errors. We fully agree that these errors affected the readability of the manuscript. In response, we have conducted a comprehensive revision of the entire manuscript to improve English fluency, eliminate redundant expressions, and correct overly literal or formal sentence structures.
All typographical issues and misspellings noted by the reviewer have been carefully corrected. The full abbreviation list has also been thoroughly checked and standardized to ensure accuracy and consistency throughout the manuscript.
Additionally, the revised manuscript has undergone extensive English editing to meet publication-level writing standards. We believe these changes have substantially improved the clarity, readability, and overall presentation of the work. We appreciate the reviewer’s constructive comments and hope the revised version addresses these concerns satisfactorily.
Comments 3: Some abbreviations are missed, including GM-CSF, PD-1, CPT1a, etc.
Response 3: We thank the reviewer for pointing this out. We have carefully reviewed the entire manuscript and corrected all missing or inconsistent abbreviations. Specifically, the abbreviations mentioned by the reviewer (GM-CSF, PD-1, CPT1a, among others) have now been defined at their first appearance and included in the abbreviation list. We have also rechecked the full text to ensure that all abbreviations are used consistently and appropriately throughout the manuscript. The updated abbreviation list is included in the revised manuscript, ensuring consistency and clarity.
Comments 4: Ensure consistent style for cytokine and gene symbols (e.g., IL-17A, not IL17A)
Response 4: Thank you for this important comment. We have thoroughly reviewed the entire manuscript and have standardized all cytokine and gene symbols to follow the appropriate nomenclature conventions. Instances such as “IL17A” have been corrected to “IL-17A,” and all similar symbols have been updated accordingly. We have also rechecked tables, figures, and the abbreviation list to ensure complete consistency throughout the manuscript.
Comments 5: Figure 1, created using BioRender, appears to have a low resolution. Please provide a high-resolution version of Figure 1.
Response 5: We thank the reviewer for pointing this out. The reduced image quality was likely due to compression when the figure was embedded in the Word document. To address this issue, we have now provided a high-resolution version of Figure 1 (exported from BioRender at publication-quality resolution) as a separate upload according to the journal’s requirements. The figure within the manuscript has also been replaced with a higher-quality version to improve clarity.
Comments 6: Add a legend explaining abbreviations for Figure 1.
Response 6: Thank you for the helpful suggestion. We have revised Figure 1 by adding a legend that defines all abbreviations used in the figure, including transforming growth factor-β (TGF-β), interleukin-15 (IL-15), interleukin-7 (IL-7), and granulocyte–macrophage colony-stimulating factor (GM-CSF). The updated figure legend is now included in the revised manuscript.
Comments 7: Please separate the “References” column in Table 1-2.
Response 7: Thank you for this helpful suggestion. In accordance with the reviewer’s comment, we have revised Tables 1 and 2 by separating the “References” column into an independent column to improve clarity and readability. We believe the updated table format is now more transparent and easier to follow.
Comments 8: It is recommended to include a figure that provides an overview of therapeutic strategies, such as repurposed drugs and targeting metabolic and microenvironmental factors.
Response 8: Thank you for this valuable suggestion. We agree that an overview of therapeutic strategies is helpful for readers. In the revised manuscript, we have already included a comprehensive summary of all therapeutic categories—including drug repurposing, metabolic interventions, residency-targeting approaches, and microenvironment-remodeling strategies—in Table 2, which clearly outlines the mechanisms, representative agents, applicable diseases, and development phases.
Given that additional figure would largely duplicate the content of Table 2, we consider the expanded table format more informative and reader-friendly, and therefore did not add an additional figure.
We hope this clarification is acceptable, and we sincerely appreciate the reviewer’s constructive input.
Comments 9: Expand limitations and discuss the lack of in vivo validation for TRM-targeting strategies.
Response 9: We appreciate the reviewer’s insightful suggestion. In response, we have expanded the discussion of limitations both in Section 5.1. Challenges for Clinical Translation and 5.2. Key Technological Directions of the manuscript. Specifically, we addressed the lack of in vivo validation for TRM-targeting strategies and other critical challenges, including:
- The heterogeneity of TRM cell populations across different tissues, complicating the development of tissue-specific targeting strategies.
- The risk of impairing protective immunity due to non-selective depletion of TRM cells, which could weaken local defenses and reduce the effectiveness of vaccines.
- The challenge of selectively silencing pathogenic TRM cell functions while preserving their protective roles in tissues.
Furthermore, we have placed additional emphasis on these limitations in Section 6 (Discussion), where we discuss the implications of the current lack of in vivo validation and the importance of addressing these gaps in future research. This section also underscores the need for in vivo models and biomarker evaluation to guide clinical trials and enable personalized therapeutic strategies.
We believe these revisions address the reviewer’s concern and provide a more comprehensive discussion of the current limitations in TRM-targeting therapies.
At last, the English language has been thoroughly revised in the updated manuscript. A native English-speaking colleague with expertise in scientific writing has carefully reviewed and polished the text to improve clarity, precision, and overall readability. We believe the revised version addresses your concern.
Reviewer 2 Report
Comments and Suggestions for Authors
Overall, this is a well written and cited review. One of my main concerns is how Trm cells are classified. The authors use Trm as a "catch-all" term, but don't specify which type of Trm cells they are talking about a majority of the time. The authors mention that some of these Trm are CD8+, but what about CD4+ and all subsets of CD4+ (TH1, TH2, TH17, TFH, Tregs, etc.)? All of those cells can be residential as well. I know CD8+ are usually the main suspects of pathology for RA, but could the authors at least have a section talking more about how CD4+ Trm could affect RA? I see that there is a section on regulatory Trm (assuming you mean Trm Tregs), but going over the other CD4+ T cell subsets (if there is enough research out there) would be good to have. I see that authors do mention some of these Trm T cells expressing some of the CD4+ phenotypes throughout the review, but it would be good if they define which subset of Trm CD4+ T cell they are talking about specifically.
Again, this is a very well written and cited review.
Author Response
Comments: Overall, this is a well written and cited review. One of my main concerns is how Trm cells are classified. The authors use Trm as a "catch-all" term, but don't specify which type of Trm cells they are talking about a majority of the time. The authors mention that some of these Trm are CD8+, but what about CD4+ and all subsets of CD4+ (TH1, TH2, TH17, TFH, Tregs, etc.)? All of those cells can be residential as well. I know CD8+ are usually the main suspects of pathology for RA, but could the authors at least have a section talking more about how CD4+ Trm could affect RA? I see that there is a section on regulatory Trm (assuming you mean Trm Tregs), but going over the other CD4+ T cell subsets (if there is enough research out there) would be good to have. I see that authors do mention some of these Trm T cells expressing some of the CD4+ phenotypes throughout the review, but it would be good if they define which subset of Trm CD4+ T cell they are talking about specifically.
Again, this is a very well written and cited review.
Response: We sincerely thank you for this valuable suggestion. The primary focus of our review is to explore the pathogenic roles of TRM cells in rheumatic autoimmune diseases, particularly their involvement in disease relapse and chronic inflammation. As such, the detailed classification of TRM cell subsets has not been the central focus. However, we recognize the importance of addressing CD4+ TRM cells and their specific role in RA, and we agree that this would further enrich the manuscript.
In response to your comment, we have expanded the Conclusion (Section 6) to include a more detailed discussion on CD4+ TRM cells in RA. Specifically, we highlight how CD4+ TRM cells, particularly TH1 and TH17 subsets, contribute significantly to RA pathology by driving synovial inflammation, enhancing cytokine secretion (e.g., IFN-γ, IL-17), and promoting tissue damage through immune interactions with fibroblasts and endothelial cells. Their persistence in the joint microenvironment is regulated by factors such as TGF-β, IL-15, and metabolic pathways like fatty acid oxidation, similar to their CD8+ counterparts.
We believe these revisions enhance the completeness of our review, and we greatly appreciate your constructive feedback. We trust these additions strengthen the manuscript and offer a more comprehensive perspective on TRM cells in RA.